# A Rare Ovarian Tumor: The Sclerosing Stromal You Do Not Expect—A Case Series in the Adolescent Population and a Literature Review

**Maria Chiara Lucchetti** [1,*], **Francesca Diomedi-Camassei** [2], **Cinzia Orazi** [3] **and Alice Tassi** [4,*]

1 Andrological Surgery Unit, Pediatric Gynecology, Department of Surgery, Bambino Gesù Children's Hospital, IRCCS (Istituto di Ricovero e Cura a Carattere Scientifico), Piazza S.Onofrio 4, 00165 Rome, Italy

2 Pathology Unit, Department of Laboratories, Bambino Gesù Children's Hospital, IRCCS (Istituto di Ricovero e Cura a Carattere Scientifico), Piazza S.Onofrio 4, 00165 Rome, Italy

3 Department of Diagnostic Imaging, Bambino Gesù Children's Hospital, IRCCS (Istituto di Ricovero e Cura a Carattere Scientifico), Piazza S.Onofrio 4, 00165 Rome, Italy

4 Clinic of Obstetrics and Gynaecology, University Hospital of Udine, P.le S.Maria Della Misericordia, 33100 Udine, Italy

\* Correspondence: mchiara.lucchetti@opbg.net (M.C.L.); alicetassi9@gmail.com (A.T.)

**Abstract:** Sclerosing stromal tumor (SST) is a rare ovarian tumor arising from the sex cord-stromal cells that occurs mainly in young adults during the second and third decades of life and rarely in pediatric and adolescent populations. The objective of this study is to report three illustrative cases of SST in young girls who had undergone surgery at our clinic in or after 2009, and to perform a literature review of this rare ovarian tumor. A retrospective chart review of female patients aged <18 years with a diagnosis of SST treated in a tertiary pediatric hospital was performed. Furthermore, a 10-year review of the SST literature was completed. Three cases of SST at our institution were outlined. After reviewing the literature, 18 SST cases were identified. The mean age at diagnosis was 13.4 years, and the reported clinical presentations were abdominal or pelvic pain and menstrual irregularity. Seven patients had abnormal hormone tests or CA-125 levels. In approximately 30% of cases, conservative surgery was performed, preserving residual ovarian tissue. In conclusion, some preoperative findings may help in suggesting the presence of SST. However, definitive diagnosis can only be made by histopathological examination. It is important to consider this tumor because, given its benign behavior, a conservative approach is preferred, particularly in this age group.

**Keywords:** sclerosing stromal tumor; pediatric; adolescent; ovarian tumor; ovarian sex cord-stromal tumor; fertility sparing

## 1. Introduction

Chalvardjian and Scully first described a sclerosing stromal tumor (SST) in 1973, identifying a new histological pattern of sex cord tumors [1]. Ovarian sex cord tumors are relatively infrequent neoplasms, accounting for approximately 8% of all primary ovarian neoplasms, and SSTs represent only 2–6% of all stromal ovarian tumors [2,3]. More than 80% of SSTs have been observed in young adults in the second and third decades of life, whereas other stromal tumors are more frequent in the fifth and sixth decades [4,5]. SST occurrence in adolescence is very rare, and even rarer cases have been reported to affect girls of premenarchal age. SST is usually unilateral, but bilateral cases have also been reported in the literature [6]. Menstrual disorders and pelvic pain are frequent manifestations at onset, as well as non-specific symptoms related to the presence of a pelvic mass [7]. Serum tumor markers are usually within the normal range, and CA-125 is seldom elevated. SSTs are mainly hormonally inactive tumors; when hormonally active, they are typically androgen secreting [8–11]. In some reported cases, typical magnetic resonance imaging (MRI) and computed tomography (CT) findings made preoperative diagnosis possible; however, due

to the high heterogeneity and rarity of SSTs, additional studies are needed to improve the knowledge of their imaging features [12]. We report three illustrative cases of SST in young girls who underwent surgery at our hospital in or after 2009. Furthermore, a literature search of the last 10 years was performed, and 18 articles were found concerning SST in pediatric and adolescent populations. Although a conservative surgical approach must be considered appropriate for these patients, it is not frequently performed because of the tumor's macroscopic features suggestive of malignancy.

## 2. Materials and Methods

A retrospective review of electronic records was performed on female patients aged 18 years and younger with a pathological diagnosis of SST between 2009 and 2021 in a tertiary pediatric hospital in Rome, Italy. Medical records were reviewed independently by two of the authors (M.C.L. and A.T.). Patients' data were collected and encrypted using Excel® database (Office 365). If any data were missing, they were logged into the database. Medical records were reviewed for age at diagnosis, presenting symptoms, blood hormone and tumor marker levels, diagnostic findings on imaging, final pathological diagnosis, treatment, and follow-up. Three cases were identified that met all the inclusion criteria (summarized in Table 1). A literature review of the PubMed and Scopus databases was performed, matching the keywords "sclerosing stromal tumor"and "ovary" and "pediatric", or "sclerosing stromal tumor" and "ovary" during the 2011–2021 period. The case reports were selected and screened by one of the authors (A.T.) and subsequently checked by another author (M.C.L.). The two authors worked independently. Case reports or case series of female patients aged <18 years with a final pathologic diagnosis of SST were included. Some of the selected cases were extrapolated by age from articles on the general population affected by SST. Only papers written in English were included in this review. The collected data consisted of personal and gynecological patient history, clinical presentation, tumor marker levels, hormonal levels, macro- and microscopic aspects of the tumor, immunohistochemistry, and surgery. The search returned 18 articles describing SST in pediatric patients, and all cases are summarized in Tables 2 and 3.

**Table 1.** Case series. List of our case series and summary of their characteristics: tumor size, laterality, age of the patient, clinical manifestations at entry, markers, blood test, macroscopic appearance, type of surgery performed, and microscopic characteristics.

| Patient | Year | Age | Laterality | Tumor Size (cm) | Clinical | Markers | Blood Exam | Gross Appearance | Surgery | Microscopically | Immunoistochemical Features | Follow-Up |
|---------|------|-----|------------|-----------------|----------|---------|------------|------------------|---------|-----------------|----------------------------|-----------|
| E.G. (Case 1) | 2009 | 13 years | Right | 8 × 7.8 × 7 | Menstrual irregularities | Normal | Normal | Solid mass | Oophorectomy | Characteristic for sclerosing stromal tumor | / | 72 months |
| S.S.A (Case 2) | 2016 | 13 years | Left | 10 × 9.5 × 9 | Menstrual irregularities | Normal | Normal | Solid mass | Salpingo-oophorectomy | Characteristic for sclerosing stromal tumor | Inhibin + vimentin + actin + desmin. +/− | 60 months |
| L.P.T.L. (Case 3) | 2018 | 13 years | Right | 3.1 × 2.8 × 2.7 | Abdominal pain Amenorrhea | / | Normal | Solid mass | Salpingo-oophorectomy | Characteristic for sclerosing stromal tumor | Inhibin + calretinin + vimentin + actin + S-100 − EMA − CD34 − | 24 months |

**Table 2.** List of the clinical cases reviewed in the literature and their characteristics: tumor size, laterality, age of the patient, clinical manifestations at entry, markers, blood test results, macroscopic appearance, type of surgery performed, and microscopic characteristics. Legend. ↑: increase in reported values.

| Author | Year | Age | Laterality | Tumor Size (cm) | Clinical Presentation | Markers | HT | Surgery | Follow-Up |
|--------|------|-----|------------|-----------------|----------------------|---------|-----|---------|-----------|
| Ahuja | 2022 | 13 years | left | 11 | Abdominal pain | ↑ CA 125 ↑ inhibin-A | ↑ T | Mass resection | 3 months |

**Table 2.** *Cont.*

| Author | Year | Age | Laterality | Tumor Size (cm) | Clinical Presentation | Markers | HT | Surgery | Follow-Up |
|---|---|---|---|---|---|---|---|---|---|
| Del Vecchio | 2020 | 17 years | Right | 4.6 × 4.1 × 4.5 | Menstrual irregularity Abnominal pain | Normal | Normal | Mass resection | 2 months |
| Chen | 2020 | 17 years | Right | 27 × 21 × 5.5 | Virilization. Amenorrhea Meig's syndrome | ↑ CA 125 | ↑ T ↑ A4 | Salpingo-oophorectomy | 22 months |
| Zhang | 2019 | 11 years | Left | 9 | Abdominal pain | Normal | Normal | Ovarian cystectomy | 60 months |
| Squillaro | 2018 | 10 months | Right | 2.7 × 2.5 × 1.7 | Precocious puberty Vaginal bleeding | Normal | Normal | Salpingo-oophorectomy | / |
| Matsutani | 2018 | 17 years | Left | 15 | Abdominal pain | ↑ CA 125 | Normal | Oophorectomy, omentectomy | / |
| Momtahan [13] | 2018 | 17 years | Right | 8 × 9 | Abdominal pain | Normal | / | Salpingo-oophorectomy | / |
| Yesil | 2016 | 17 years | Left | 5 × 4 | Abdominal pain | / | / | Paraovarian mass resection | / |
| Naidu | 2015 | 14 years | Bilateral | 11 × 9 × 8 | Primary amenorrhea | Normal | ↑ T ↑ 17OHP | Left salpingo-oophorectomy - right ovarian cystectomy | 3 months |
| Atram | 2014 | 15 years | Right | 8 × 5 × 3 | Mestrual irregularity Pelvic pain | / | / | / | / |
| Chaurasia | 2014 | 7 years | Right | 16 × 12.5 × 10 | Precocious puberty Vaginal bleeding | Normal | ↑ E2 | Salpingo-oophorectomy | 60 months |
| Yen | 2014 | 9 years | Left | 15 × 8.5 × 6 | Virilization | Normal | ↑ T ↑ A4 ↑ 17OHP ↑ DHEAS | Salpingo-oophorectomy | 2 months |
| Limaiem | 2013 | 16 years | Left | 15 × 11 × 7 | Mestrual irregularity Pelvic pain | Normal | Normal | Salpingo-oophorectomy | / |
| Mahadevappa | 2012 | 16 years | Left | 17 × 13 × 5 | Mestrual irregularity Abdominal mass Meig's syndrome | ↑ CA 125 | Normal | Mass resection | / |
| Duzcu | 2012 | 17 years | / | 7.5 | Mestrual irregularity Menometrorrhagia | Normal | / | / | / |
| Dilbaz [14] | 2011 | 14 years | Right | 8 | Dysmenorrhea Pelvic pain | Normal | Normal | Mass resection | / |
| Onur | 2011 | 12 years | Right | 5 | Mestrual irregularity Abdominal pain | Normal | Normal | Salpingo-oophorectomy | / |
| Park | 2011 | 11 years | Left | 9 | Virilization | Normal | ↑ T ↑ A4 ↑ 17OHP ↑ DHEAS | Oophorectomy | 6 month |

**Table 3.** List of the clinical cases reviewed in the literature and their characteristics: tumor size, laterality, age of the patient, clinical manifestations at entry, markers, blood test results, macroscopic appearance, type of surgery performed, and microscopic characteristics.

| Author | Year | Gross Appearence | Microscopically | Immunoistochemical Features |
|---|---|---|---|---|
| Ahuja | 2022 | Tan-yellow solid mass | pseudolobular pattern, hypercellular and hypocellular myxoid areas with prominent, branching vasculature. Luteinized cells and occasional interspersed spindled cells were noted | / |
| Del Vecchio | 2020 | Solid mass | Pseudolobular pattern alternating hypocellular and hypercellular areas, the presence of luteinized theca-like cells with vacuolated cytoplasm and fusiform fibroblasts-like cells, fibrosis and oedematous stroma | inhibin + calretinin + actin + Ki67 < 10% |
| Chen | 2020 | Cystic and solid, encapsulated mass | Pseudolobular pattern, round and short spindle cells were predominant. | inhibin + calretinin + CD99 + SMA − EMA − CK − Desmin − Ki67 3% |
| Zhang | 2019 | Cystic and solid, encapsulated mass | Pseudolobular pattern, theca-like cells with eccentric nuclei, spindle-shaped fibroblast-like cells with elongated nuclei | inhibin + calretinin + vimentin + CD99 + SMA + CD34 + Ki67 15% |
| Squillaro | 2018 | Tan-pink, smooth, and glistening surface | Multiple nodules with central hyalinization and scattered degenerative vacuolated cells surrounded by fibroblast-like cells, resembling corpora albicantia-like appearance | inhibin + calretinin + SMA − CD34 − |
| Matsutani | 2018 | Cystic mass | Pseudolobular pattern, collagen-producing bland spindled cells and rounded epithelioid cells | inhibin + |
| Momtahan [13] | 2018 | Dermoid cyst like | / | / |
| Yesil | 2016 | Cystic mass | Pseudolobular pattern in which cellular spindle cell zones alternated with edematous and collagenous hypocellular zones | inhibin + calretinin + CD99+ SMA + Desmin − Caldesmon − Ki67 10% |
| Naidu | 2015 | Left: solid encapsulated mass with focal calcification Right: solid lobulated mass | Pseudolobular pattern, collagen-producing spindled cells and hypocellular areas with focally edematous and fibrous stroma | / |
| Atram | 2014 | Ovarian torsion, solid, cystic mass | Pseudolobular pattern, spindle shaped and round to oval cells with vesicular nuclei and a moderate amount of eosinophilic cytoplasm | / |
| Chaurasia | 2014 | Encapsulated ovarian mass | Pseudolobular pattern, spindle and round vacuolated clear cells. | inhibin + vimentin + SMA + CK − |
| Yen | 2014 | Solid mass | Spindle cells, with elongated nuclei with pointy ends and scant cytoplasm. Hypercellular areas: cells with vacuolated or eosinophilic cytoplasm, round nuclei and small nucleoli | inhibin + calretinin + vimentin + CD34 − EMA − |
| Limaiem | 2013 | Solid mass | Pseudolobular pattern, oedematous and collagenous areas, spindle-shaped cells | inhibin + vimentin + SMA + CK − |
| Mahadevappa | 2012 | Solid mass | Pseudolobular pattern. Hypercellular areas: spindle-shaped cells, polygonal tumor cellsmyoid cells. Residual ovarian tissue | / |
| Duzcu | 2012 | Solid mass | Cell areas with vacuoles and cytoplasm with prominent nuclei. Presence of round-oval shaped cells and spindle cells. | inhibin + calretinin + vimentin + ER − PR + SMA + AFP − EMA − CK − |
| Dilbaz [14] | 2011 | Solid mass | Pseudolobular pattern, vacuolated spindle and polygonal cells | / |

**Table 3.** *Cont.*

| Author | Year | Gross Appearence | Microscopically | Immunoistochemical Features |
|---|---|---|---|---|
| Onur | 2011 | Solid mass | Cellular areas and edematous and hyalinized stromal elements. Cellular areas included spindle-shaped fibroblasts and polygonal cells with vacuolated cytoplasm | inhibin + vimentin + |
| Park | 2011 | Solid mass | Fibroblasts, rounded vacuolated cells and prominent thin walled vessels, edematous and collagenous hypocellular areas | inhibin + vimentin + SMA + S100 − CK − |

### 3. Institutional Cases and Literature Review

*3.1. Case 1*

A 13-year-old girl presented to our attention with an ultrasound finding of a pelvic mass with a history of menstrual irregularities that began a few months earlier, at the age of 12 years. She had no relevant medical or family history. Her antenatal, perinatal, and developmental histories were unremarkable. Abdominal and physical examinations were benign. There were no abnormal results in routine blood, liver, or kidney function tests. Levels of tumor markers (alpha-fetoprotein, CEA, CA-125, beta-hCG, and LDH) were within normal limits. Hormonal levels were not evaluated, given the normal pubertal development and the absence of clinical signs of hyperandrogenism. Pelvic ultrasound scan was repeated at our hospital and confirmed the presence of a large, mixed-solid and cystic, partly microcalcified, right pelvic mass measuring 8 × 7.8 × 7 cm, displacing the uterus. Pelvic MRI showed an uneven expansive solid and cystic formation in the right paramedian pelvic region, which caused compressive effects on nearby structures and displaced the uterus (Figure 1A–C). During surgery, a laparotomic approach revealed an enlarged right ovary completely occupied by a solid formation. No residual ovarian tissue was identifiable, and right salpingo-oophorectomy was performed. Histology revealed a predominantly solid neoplasm, with a proliferation of spindle and round cells (Figure 2A) separated in lobules by a richly vascularized stroma. The stroma showed both edematous and fibrous areas. The microscopic diagnosis was sclerosing stromal tumor. Postoperative recovery was uneventful, and the patient was discharged on the fourth postoperative day. The patient was subjected to periodic ultrasounds for 12 months. At the last follow-up 72 months after surgery, there were no signs of recurrence.

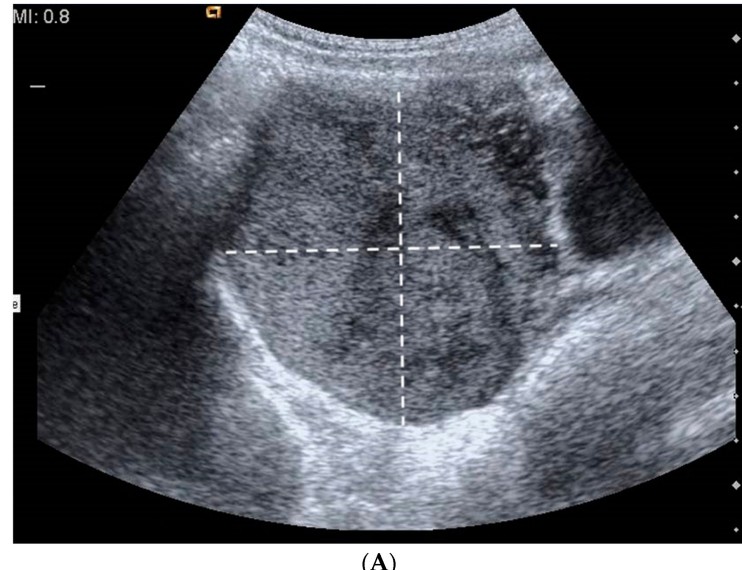

(**A**)

**Figure 1.** *Cont.*

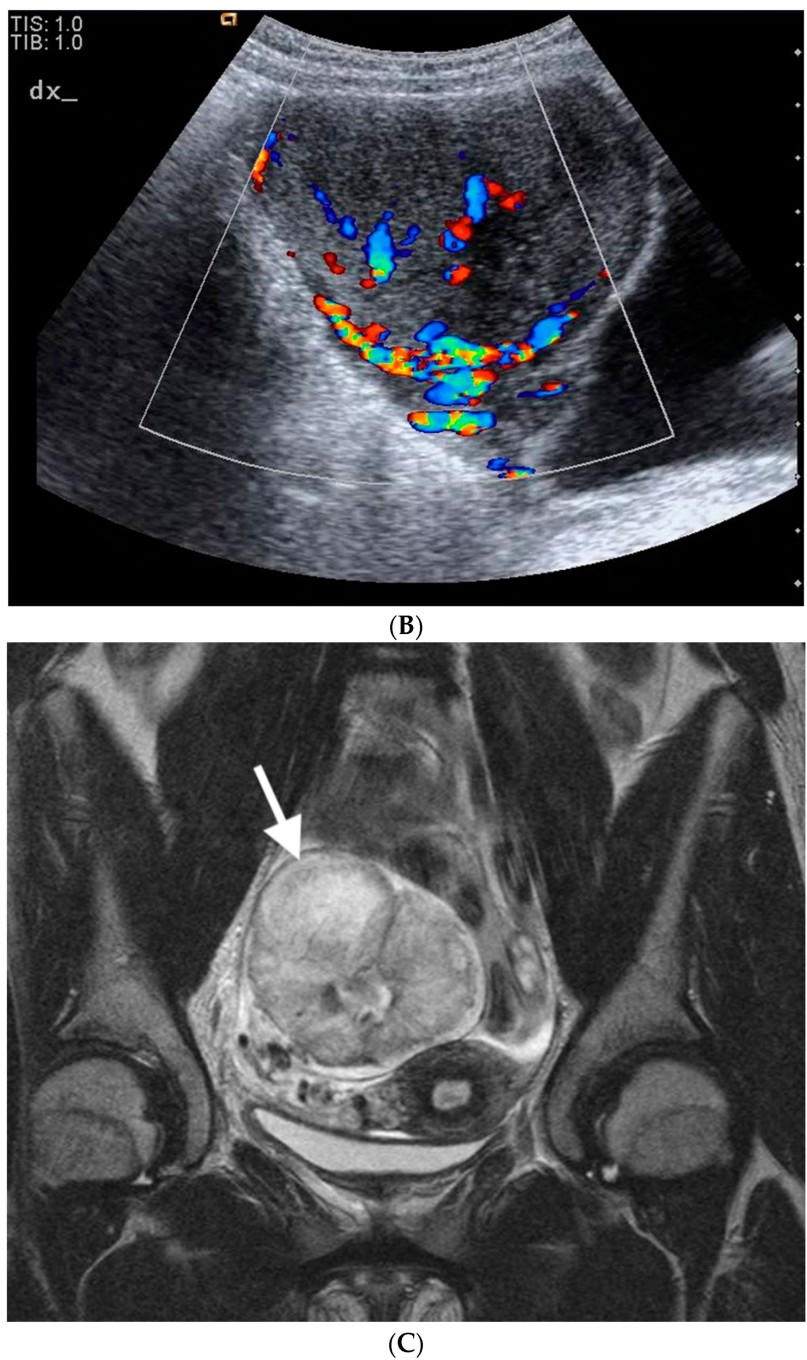

**Figure 1.** (**A**) Ultrasound image reveals a solid mass with focal hypoechoic components, possibly due to necrosis; (**B**) Doppler US image reveals prominent peripheral vasculature; (**C**) MR images show an uneven expansive mass in the right paramedian pelvic region, which dislocates the uterus. The arrow indicates the formation found on the MRI.

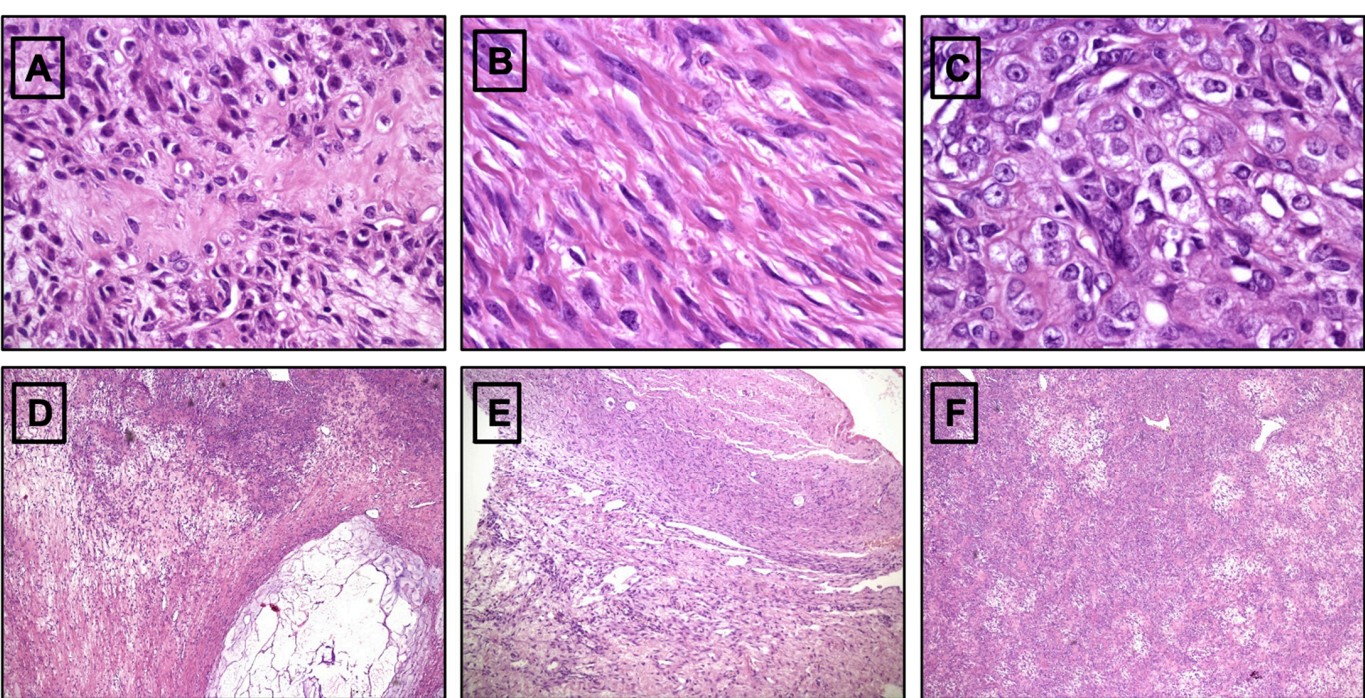

**Figure 2.** (**A**) Spindle and round cells intermingled together in a hyaline stroma (asterisk) (HE, 40×); (**B**) Fascicular spindle cell proliferation (HE, 63×); (**C**) Proliferation of round, epithelioid clear cells (arrowhead) (HE 63×); (**D**) A clear gelatinous area is observed (arrow) (HE, 5×); (**E**) A rim of residual ovarian parenchyma is shown in the upper part of the picture (rectangle) (HE, 10×); (**F**) Alternation of hyper- and hypocellular areas in the same tumor ("dark" and "clear" proliferation pattern) (HE, 5×).

### 3.2. Case 2

A 13-year-old girl presented with an ultrasound finding of a pelvic mass and a recent menstrual history of irregularities. The patient reported menarche at age 11, menstrual cycles initially regular but irregular later on. She had no relevant medical or family history. Her antenatal, perinatal, and developmental histories were unremarkable. Upon physical examination, the mass was palpable in the left iliac fossa. There were no abnormal results in routine blood, liver, or kidney function tests. Levels of tumor markers (alpha-fetoprotein, CEA, CA-125, beta-hCG, and LDH) were within normal limits. Pelvic ultrasound confirmed irregular growth of the left ovary measuring 10 × 9.5 × 9.0 cm. Pelvic MRI showed an uneven expansive solid formation in the left paramedian pelvic region of approximately 10 × 10 cm, which caused compressive effects on nearby structures and dislocated the uterus. Laparotomy revealed an enlarged left ovary completely occupied by a solid parenchymatous mass. No residual ovarian tissue was detected, and a left salpingo-oophorectomy was performed. Histological examination revealed spindle cell proliferation (Figure 2B) admixed with a component of rounded, epithelioid clear cells (Figure 2C). The mass was divided into lobules by hyaline and richly vascularized stroma that contained gelatinous (Figure 2D) and mixoyd areas. A thin line of ovarian parenchyma was recognizable in the periphery (Figure 2E). The pathology report was consistent with that of SST. Postoperative recovery was uneventful, and the patient was discharged on the fourth postoperative day. The patient was subjected to periodic ultrasounds for 12 months. At the last follow-up 60 months after surgery, there were no signs of recurrence.

### 3.3. Case 3

A 13-year-old girl presented to the emergency room with abdominal pain and amenorrhea. Menarche is reported at the age of 11 years, with regular menstruation until about a year before, when amenorrhea started. The patient also reported weight loss in the previous year. During the observation period, a transabdominal pelvic ultrasound scan

was performed, which revealed a globular mass in the right ovary and a suspicion of an arcuate uterus. She had no relevant medical or family history. Her antenatal, perinatal, and developmental histories were unremarkable. Abdominal and physical examinations were benign. There were no abnormal results in routine blood, liver, or kidney function tests. No tumor markers were identified. A new ultrasound study demonstrated a right-side solid adnexal mass, subsequently confirmed by MRI, measuring $3.1 \times 2.8 \times 2.7$ cm. Laparoscopy revealed an enlarged right ovary, with peripherally displaced residual ovarian tissue. The mass was removed, leaving a consistent amount of residual ovarian tissue. Microscopically, proliferation of elongated cells growing in hypercellular or hypocellular areas was observed (Figure 2F). Intermixed lobules with oval or rounded clear cells were observed. Hyalin and mixoyd vascular stroma appeared dividing the neoplasm in broad lobules. Histology was consistent with a sclerosing stromal tumor with positive resection margins. The postoperative hospitalization was uneventful, and the patient was discharged on the third day. However, considering the pathology report, a second-look surgery was performed two months later, which consisted in a laparoscopic right salpingo-oophorectomy. Pathology described a normal parenchymal ovarian tissue with numerous follicles in various maturative phases and a small nodule of the previous pathology (sclerosing stromal tumor), surrounded by lymphomonocytic, plasmacellular, and gigantocellular inflammatory infiltrating tissue by a foreign body. The fallopian tube had a preserved and slightly congested architecture. Postoperative recovery was uneventful, and the patient was discharged on the third day. The patient underwent periodic ultrasound examinations for 12 months. At the last follow-up 24 months after surgery, there were no signs of recurrence.

*3.4. Literature Review*

Within the published English literature, we were able to find 18 cases of SST under the age of 18 years over a 10-year period. All published cases and data are presented in Tables 2 and 3, respectively. The mean age at diagnosis was 13.4 years (range, 10 months–17 years). The affected ovary was the right ovary in 9 cases and the left ovary in 7; there was one bilateral case, and laterality was not reported in one case. The mean tumor size at diagnosis was 10.8 cm (range, 2.7–27.0 cm). An increase in serum CA-125 levels was reported in 4 cases, while the majority of patients had normal serum levels when tested for this marker (12 patients). At presentation, among six checked cases, elevated sexual hormone levels were present in only one case, represented by elevated estrogen levels; however, in most cases, sexual hormone levels were not checked. The main presenting symptoms were abdominal or pelvic pain (10 cases), menstrual irregularity (8 cases, 2 of which were amenorrhea), clinical signs of virilization (3 cases), Meig's syndrome (2 cases), and precocious puberty (2 cases). In some cases, more than one symptom was present. Conservative surgery, with preservation of residual ovarian tissue, was performed in six patients, while a salpingo-oophorectomy was performed in 10 patients. In two cases, the type of surgical treatment was not specified. On immunohistochemical investigation, inhibin was present in 70.6% ($n = 12$), calretin in 47.1% ($n = 8$), vimentin in 41.2% ($n = 7$), and smooth muscle actin (SMA) in 29.4% ($n = 5$) of cases. The mean follow-up period after surgery, reported only in selected cases, was 22.1 months (range, 2–60 months).

## 4. Discussion

Ovarian neoplasms are rare in the pediatric and adolescent populations; the annual incidence is approximately 2.2 cases/100,000, and about 25% of them are malignant, being mainly tumors of germ cell origin [15,16]. Ovarian sex cord-stromal tumors (OSCTs) are a group of tumors arising from non-germ-cell components of ovarian tissue, including granulosa cell tumors, fibrothecomas, Sertoli–Leydig cell tumors, steroid cell tumors, and sclerosing stromal tumors [3]. It has been historically reported that they represent 8–13% of all primary ovarian neoplasms, with SSTs accounting for only 2–6% of all OSCTs [2,3]. At our institution, we observed, in the same study period and in a population strictly under 18 years of age, 67 primary ovarian tumors. Most of them ($n = 38$) were germinal tumors

(56.7%), five of them were immature teratomas, 21 (31.3%) were epithelial tumors (3 of them borderline), and 8 (11.9%) were sex cord-stromal tumors. Of these, three were granulosa cell tumors (two, juvenile type), one was a leiomyoma, one was a Sertoli–Leydig, and three were SSTs. SSTs are thought to originate from elements of the theca externa, namely the perifollicular myoid stromal cells, a population of muscle-specific actin-positive cells or from a pre-existing ovarian fibroma [17–19]. Other authors have recognized a close relationship between thecomas and SSTs, as they share some morphological features and antigenic determinants such as smooth muscle actin and vimentin [20]. It can then be assumed that SSTs may arise from pluripotent immature stromal cells of the ovarian cortex, along with other similar elements [3]. More than 80% of SSTs occur in young adults in the second and third decades of life (mean age 27 years) [2], and the majority of cases are unilateral, even though bilateral cases have rarely been reported [4–6]. In the considered literature, based merely on the pediatric and adolescent populations, the mean age was 13.4 years. Similarly to the adult population, most SSTs were unilateral, with only three described bilateral lesions [6,21,22]. In our case series, we reported a mean age of 13.0 years, and all the cases were unilateral. We also considered a recent article by Devins et al., who dealt with the topic of SST very broadly, emphasizing the difficulty of a differential diagnosis that includes both benign and malignant pathologies. The article includes 100 cases of SST that also affect pediatric patients. Articles in the literature concerning the cases of SST among pediatric patients in the last ten years are also reported in our manuscript [23]. The typical clinical presentation of SSTs in the pediatric and adolescent populations is pelvic or abdominal pain and non-specific symptoms related to the presence of a pelvic mass. Serum tumor markers, such as inhibin A, human chorionic gonadotropin, alpha-fetoprotein, and lactate dehydrogenase, are generally within normal limits. Sometimes, CA-125 can be elevated; among the analyzed articles, less than 20% showed an increased level of this marker [9–11,24–27]. After surgery, serum CA-125 levels returned to normal in most of the reported cases. None of the reported cases (Table 1) showed an increase in serum CA-125 levels when tested. Some clinical features may be associated with Meig's syndrome (benign ovarian tumor, hydrothorax, and ascites). Two similar cases have been reported in the pediatric literature, with SST being the ovarian tumor [26,27], but there are none included among our cases (Tables 1 and 2). SST was initially reported as a hormonally inactive benign ovarian tumor. However, in 1975, Damjanov et al. reported the production of steroid hormones by SSTs [28]. Androgens and, less frequently, estrogens can lead to menstrual disturbances and signs of virilization or precocious puberty, as reported in several studies [8,12,21,25–27,29–34]. Menstrual irregularities are very common in the first two years after menarche; therefore, this symptom may be underestimated in this population or simply attributed to a physiologically abnormal hormonal production. None of our cases showed hormonal alterations; however, menstrual irregularities were observed in all cases. Ultrasound findings revealed no characteristic features for an accurate differentiation of SSTs from other malignant or benign ovarian masses. Ovarian SSTs can show a mixed pattern with cystic and solid components, and a marked vascularity, similar to most malignant ovarian tumors. Adequate preoperative evaluation is essential, especially in young patients who can benefit the most from ovarian preservation; however, SST imaging is not immediate. It is difficult to differentiate SSTs from malignant masses. Some predictive models, such as the IOTA system, can be useful for predicting the malignancy or benignity of an ovarian mass [35]. According to a 2021 study [35], the diagnostic accuracy of the ADNEX IOTA model for sex cord-stromal tumors in distinguishing malignant from benign tumors was 84%. Another radiological indicator of benignity is the 'ovarian crescent sign'. This finding is consistent with the presence of a rim of healthy ovarian tissue in the ipsilateral ovary in the presence of an ovarian mass. The absence of a crescent sign is associated with malignancy, high sensitivity and specificity [36–38]. Studies in the pediatric population have also confirmed this finding, although the crescent sign was less predictive in premenarchal patients [38]. According to a 2019 systematic review [39] of predictive models of malignancy, the ovarian crescent sign, along with preoperative tumor markers, is

a useful tool for distinguishing benign from malignant lesions [40]. Additional information may be obtained with other imaging techniques such as CT or MRI. Imaging results included a large mass with hyperintense cystic components or a heterogeneous solid mass of intermediate to high signal intensity on T2 MRI. The thick peripheral hypointense border on T2-weighted imaging indicates a compressed ovarian cortex due to a slow-growing tumor. The pattern of contrast enhancement of ovarian SST on dynamic CT and MRI is characteristic and results in an improvement in peripheral contrast in the initial phase after administration of the contrast medium, followed by a centripetal progression in the delayed phase. Early enhancement reflects cellular areas with prominent vascular networks, and an area of prolonged enhancement in the inner part of the mass represents the hypocellular area of collagen. In contrast, most malignant ovarian neoplasms show early improvement and redness of the contrast medium [12,41]. Furthermore, according to previous reports, ovarian SSTs show elasticity similar to that of the uterine myometrium in ultrasonic elastography [12]. The wide variety of imaging presentations among cases of SST and their rarity require further study to increase the diagnostic rates of these clinical tools.

In our reported cases, imaging always showed a solid ovarian mass with rare cystic components (Figures 1 and 2). In reported cases 1 and 2, the suspicion of malignancy arose from the large dimensions of the neoformations (Case 1: 8 × 7.8 × 7 cm; Case 2: 10 × 9.5 × 9 cm), their solid or partially solid and cystic appearance (in case 1 also with a calcified component), and the absence of the 'crescent sign'. In reported case 3, on the basis of the small dimensions of the mass (3.1 × 2.8 × 2.7 cm), we opted for conservative surgical therapy. Subsequently, a radical intervention was necessary due to the presence of local residual disease. In none of the 3 cases was the 'contrast distribution' described in the literature evaluated. SST is a benign, stromal tumor that typically exhibits a pseudolobular appearance with broad streaks of fibrous stroma separating tumor nodules. In our cases, dilated and branching thin-walled vessels were characteristic. The cellular component displayed spindled, rounded, and weakly luteinized cells. SST typically has low mitotic activity, although rare cases with high mitotic activity have been reported in the literature [42]. In a single case, an SST with marked atypia mimicking an undifferentiated sarcoma has been reported [43]. In some cases, differential diagnosis between sclerosing stromal tumors and juvenile granulosa cell tumors with pronounced stromal sclerosis may be difficult. However, the characteristic vascular pattern and low mitotic activity can help to clarify the diagnosis [43,44]. Rarely, vacuolated cells present in sclerosing stromal tumors may have a signet ring-like structure that mimics Krukenberg's tumor. However, the latter is mostly bilateral, lacks the characteristic pseudolobular pattern of sclerosing stromal tumors, and usually occurs during the sixth and seventh decades of life. Furthermore, Krukenberg's tumor signet ring-like proteins contain mucin rather than lipids, and show nuclear atypia and mitotic activity [19,34]. Sclerosing stromal tumor cells were mainly positive for vimentin, SMA, and inhibin, strongly suggesting their stromal origin. Other markers, such as calretinin or desmin, can be positive or negative, whereas epithelial and S-100 markers are negative. However, other authors have stated that inhibin and calretinin are the most useful markers for distinguishing stromal tumors [6,19,45]. In the tumors observed in our hospital (Table 1) by the same pathologist, histological features met the criteria for the diagnosis of sclerosing stromal tumors described in the "WHO Classification of Tumours of Female Reproductive Organs" [46] and in the "AFIP Atlas of Tumor Pathology" (*Tumors of the ovaries, maldeveloped gonads, fallopian tube, and broad ligament*) [47]. In 2017, Park et al. reported the overexpression of TFE3 in sclerosing stromal tumors, even if the underlying mechanism is unknown and further investigation is needed [48]. Differential diagnosis includes other sex cord-stromal tumors, such as fibroma, thecoma, lipid cell tumor, vascular tumors, malignant tumors (Krukenberg's tumor), and non-neoplastic conditions, such as massive ovarian edema. Pseudolobulation, prominent vessels, and lutein cells and fibroblasts admixed in a jumbled manner are the main features that should prompt the consideration of a benign diagnosis, differentiating SSTs from other ovarian neoplasms. Comparing the pediatric population with SST and the adult one, the most common symp-

toms are pelvic or abdominal pain, and menstrual irregularities in both [7,23]. An article by Devins et al.[21] reports pleural effusion as a symptom in the general population, but this was not present in our cases nor our, or others', review [7]. The macroscopic aspect of the SST of the pediatric population overlaps with the data reported in the literature of the general population. Finally, also the immunohistochemistry of the pediatric population is comparable to the data reported in the adult population [7,23]. All ovarian SSTs reported in the literature were benign and were successfully treated with ovarian cystectomy or unilateral oophorectomy. Unfortunately, conservative surgery with preservation of ovarian tissue was performed in less than 30% of the reported cases due to concern for malignancy. To avoid unnecessary oophorectomy and preserve as much ovarian tissue as possible, the best approach should be conservative 'until proven otherwise'. Prophylactic salpingectomy as a strategy for reducing the risk of ovarian cancer is indicated at the time of oophorectomy. The 2004 COG surgical staging guidelines (written for ovarian primary germ cell tumors) [49] are always applied to lower the oncological risk, but only a frozen section (if available) or a definitive diagnosis of malignancy should prompt ablative surgery. When staging guidelines have been correctly applied to minimize the risk of tumor spreading, a second-look procedure can be safely performed in case of malignancy or affected resection margins. Although it is commonly believed that the reproductive potential of women who have undergone unilateral oophorectomy is similar to that of normal women, several studies have reported the requirement for higher gonadotrophin doses and the retrieval of fewer oocytes at ovum pick-up in women with previous unilateral oophorectomy when submitted to assisted reproductive techniques [50,51]. Furthermore, the age at onset of menopause of women with two ovaries differs by approximately one year from the age of onset of menopause in women who have undergone a unilateral oophorectomy [52,53].

## 5. Conclusions

Sclerosing stromal tumors are benign ovarian neoplasms. They are rare, especially in the pediatric and adolescent populations. Owing to the variety of presentations, obtaining a preoperative diagnosis is often difficult. However, SSTs should be considered in all young women with ovarian masses and lack of malignant features. A conservative, ovary-sparing surgical approach has to be considered mandatory in all patients with SST, especially in younger patients with a long life and fertility expectancy. Longer and more accurate follow-up studies are needed to determine the consequences of surgery for these tumors. The differential diagnosis of SSTs is broad, including fibromas, thecomas, solitary fibrous tumors, pregnancy luteomas, myxomas, other ovarian sex cord-stromal tumors with sclerosis and, rarely, Krukenberg's tumors. Strict adherence to the requirement of pseudolobulation, prominent (usually ectatic) vessels, and lutein cells and fibroblasts admixed in a jumbled manner will distinguish the neoplasm from others in the differential.

**Author Contributions:** Conceptualization, M.C.L., C.O., F.D.-C. and A.T.; Validation, C.O.; Original Draft Preparation, A.T., M.C.L. and F.D.-C.; Writing—Review and Editing, A.T. and M.C.L.; Supervision, M.C.L. All authors have read and agreed to the published version of the manuscript.

**Funding:** This research received no external funding.

**Institutional Review Board Statement:** Not applicable.

**Informed Consent Statement:** Not applicable.

**Data Availability Statement:** Data available on request due to restrictions eg privacy or ethical.

**Conflicts of Interest:** The authors declare no conflict of interest.

## Abbreviations

17OHP—17$\alpha$-Hydroxyprogesterone; A4—androstenedione; DHEAS—dehydroepiandrosterone sulfate; E2—estradiol; HT—hormonal tests; T—testosterone.

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
