# Peer review of "A Rare Ovarian Tumor: The Sclerosing Stromal You Do Not Expect—A Case Series in the Adolescent Population and a Literature Review"

_pediatrrep, doi:10.3390/pediatric15010004_

Round 1

Reviewer 1 Report

Brief summary  Ovarian neoplasms are a rare occurrence in female children and adolescents, and 90% of such masses are proven benign. SSTs of the ovary are extremely rare benign neoplasms with atypical presentations that makes the preop diagnosis challenging. Although examination by ultrasound is the preferred auxiliary in the diagnosis of ovarian pathology, it could not distinguish between malignant and benign tumors.     General concept comments:   COMMENT N1: You describe a 'typical contrast enhancement pattern' of these tumors (improvement in peripheral contrast soon after administration of the contrast medium, followed by centripetal progression in the delayed phase). The reported MRI description is limited to anatomical details, and appears poor in defining the patterns of vascularization : -Did you analyze the contrast distribution as reported in the literature? - In your patients, which MRI detail raised the suspicion of malignancy and justified the radical surgical procedure?
      COMMENT N.2 Due to concern for malignancy,  in your series conservative surgery with preservation of ovarian tissue was performed in less than 30% of cases . You state that to avoid unnecessary oophorectomy and preserve as much ovarian tissue as possible, the best approach should be conservative ‘until proven otherwise’.   Salpingectomy is a strategy for ovarian cancer risk reduction in women. Which Indications (clinical or radiological) do you apply for opportunistic salpingectomy in adolescents ?     COMMENT N.3 The laparoscopic approach should be considered as a feasible option, especially in cases where there are no predictive factors for a malign disease. The most severe complication encountered in laparoscopic surgery are associated with cystic lesions and consists in the rupture of a cyst of unknown nature.    Can you explain the reasons for choosing laparotomy?       Specific comments:   Line 286 and 287: SST typically exhibits a pseudolobular appearance with broad streaks of fibrous stroma separating tumor nodules. SST is a benign, stromal tumor that typically exhibits a pseudolobular appearance with broad streaks of fibrous stroma separating tumor nodules
The sentence is repeated  

Author Response

Did you analyze the contrast distribution as reported in the literature? 

Our answer has been inserted in the text within the discussion.

In your patients, which MRI detail raised the suspicion of malignancy and justified the radical surgical procedure?

Our answer has been inserted in the text within the discussion.

Salpingectomy is a strategy for ovarian cancer risk reduction in women. Which Indications (clinical or radiological) do you apply for opportunistic salpingectomy in adolescents?     

Prophylactic salpingectomy, as a strategy for reducing the risk of ovarian cancer, is indicated at the time of oophorectomy

The laparoscopic approach should be considered as a feasible option, especially in cases where there are no predictive factors for a malign disease. The most severe complication encountered in laparoscopic surgery are associated with cystic lesions and consists in the rupture of a cyst of unknown nature.    Can you explain the reasons for choosing laparotomy? 

A laparotomic approach is preferred in cases with a higher probability of malignancy.      

Specific comments:   Line 286 and 287: SST typically exhibits a pseudolobular appearance with broad streaks of fibrous stroma separating tumor nodules. SST is a benign, stromal tumor that typically exhibits a pseudolobular appearance with broad streaks of fibrous stroma separating tumor nodules
The sentence is repeated  

We corrected it along with your suggestion

Reviewer 2 Report

The authors present three cases of ovarian sclerosing stromal tumor (SST) in the adolescent population along with review of the literature. The authors conclude that SSTs should be a diagnostic consideration in this population, particularly as it allows for a more conservative management.

Some strengths of the study include its novelty, as there is limited review of SSTs in the pediatric population. The authors also summarize the reported characteristics of SSTs in the pediatric population, including imaging, histology, and immunohistochemistry findings.

Minor revision:

1. There are a significant number of misspells and grammatical errors that distract from the manuscript (e.g. Table 1: "casw" review; "pathgological" in Materials and Methods, "appearence" in table headings, etc).

2. Are the findings in this population (imaging, gross, histoloy, and immunohistochemistry) comparable to SSTs in the adult population?

3. As the authors advocate conservative surgery in cases of SSTs, which they found was performed in less than 30% of cases due to concern for malignancy, it would be beneficial to highlight findings during frozen section that should prompt the consideration of a benign diagnosis.

Author Response

There are a significant number of misspells and grammatical errors that distract from the manuscript (e.g. Table 1: "casw" review; "pathgological" in Materials and Methods, "appearence" in table headings, etc).

We corrected them

Are the findings in this population (imaging, gross, histoloy, and immunohistochemistry) comparable to SSTs in the adult population?

Comparing the pediatric population with SST and the adult one: in both the most common symptoms are pelvic or abdominal pain, and menstrual irregularities. An article by Devins et al. reports pleural effusion as a symptom in the general population, not present in our cases and in our or other reviews. The macroscopic aspect of the SST of the pediatric population overlaps with the data reported in the literature of the general population. Finally, also the immunohistochemistry of the pediatric population is comparable to the data reported in the adult population.

As the authors advocate conservative surgery in cases of SSTs, which they found was performed in less than 30% of cases due to concern for malignancy, it would be beneficial to highlight findings during frozen section that should prompt the consideration of a benign diagnosis.

We added it to the manuscript: “pseudolobulation, promiment vessels, and lutein cells and fibroblasts admixed in a jumbled manner are the main features that should prompt the consideration of a benign diagnosis, differetiating SST from other ovarian neoplasms.”